# Managing Gut Microbiota through In Ovo Nutrition Influences Early-Life Programming in Broiler Chickens

**DOI:** 10.3390/ani11123491

**Published:** 2021-12-07

**Authors:** Abdelrazeq M. Shehata, Vinod K. Paswan, Youssef A. Attia, Abdel-Moneim Eid Abdel-Moneim, Mohammed Sh. Abougabal, Mohamed Sharaf, Reda Elmazoudy, Wejdan T. Alghafari, Mohamed A. Osman, Mayada R. Farag, Mahmoud Alagawany

**Affiliations:** 1Department of Dairy Science and Food Technology, Institute of Agricultural Sciences, Banaras Hindu University, Varanasi 221005, India; vkpaswan.vet@gmail.com; 2Department of Animal Production, Faculty of Agriculture, Al-Azhar University, Cairo 11651, Egypt; dr.mabougabal@azhar.edu.eg; 3Agriculture Department, Faculty of Environmental Sciences, King Abdulaziz University, Jeddah 21589, Saudi Arabia; 4Nuclear Research Center, Biological Applications Department, Egyptian Atomic Energy Authority, Abu-Zaabal 13759, Egypt; aeabdelmoneim@gmail.com; 5Department of Biochemistry and Molecular Biology, College of Marine Life Sciences, Ocean University of China, Qingdao 266003, China; mohamedkamel@azhar.edu.eg; 6Department of Biochemistry, Faculty of Agriculture, Al-Azhar University, Cairo 11651, Egypt; 7Biology Department, College of Science, Imam Abdulrahman Bin Faisal University, P.O. Box 1982, Dammam 31441, Saudi Arabia; rhelmazoudy@iau.edu.sa (R.E.); maaosman@iau.edu.sa (M.A.O.); 8Basic and Applied Scientific Research Center, Imam Abdulrahman Bin Faisal University, P.O. Box 1982, Dammam 31441, Saudi Arabia; 9Clinical Nutrition Department, Faculty of Applied Medical Sciences, King Abdulaziz University, Jeddah 21589, Saudi Arabia; Walghefari@kau.edu.sa; 10Forensic Medicine and Toxicology Department, Faculty of Veterinary Medicine, Zagazig University, Zagazig 44519, Egypt; dr.mayadarf@gmail.com; 11Poultry Department, Agriculture Faculty, Zagazig University, Zagazig 44519, Egypt

**Keywords:** gut microbiota, in ovo, probiotics, prebiotics, early-life programming

## Abstract

**Simple Summary:**

Modulating the gut microbiota has been proposed as a potential strategy for improving host health and productivity and avoiding the undesirable effects on gut health and the immune system. In ovo technology, through which a tiny quantity of material is injected into the bird’s egg/embryo during incubation, offers a novel alternative for delivering bioactive compounds to an embryo before hatching. Recent experiments from various researchers showed the benefits of in ovo feeding technology on the chicken body weight, feed conversion ratio, and pectoral meat yield. Using early-life programming via the in ovo technique with different feed additives may be possible to avoid selected the metabolic disorders, poor immunity, and pathogen resistance.

**Abstract:**

The chicken gut is the habitat to trillions of microorganisms that affect physiological functions and immune status through metabolic activities and host interaction. Gut microbiota research previously focused on inflammation; however, it is now clear that these microbial communities play an essential role in maintaining normal homeostatic conditions by regulating the immune system. In addition, the microbiota helps reduce and prevent pathogen colonization of the gut via the mechanism of competitive exclusion and the synthesis of bactericidal molecules. Under commercial conditions, newly hatched chicks have access to feed after 36–72 h of hatching due to the hatch window and routine hatchery practices. This delay adversely affects the potential inoculation of the healthy microbiota and impairs the development and maturation of muscle, the immune system, and the gastrointestinal tract (GIT). Modulating the gut microbiota has been proposed as a potential strategy for improving host health and productivity and avoiding undesirable effects on gut health and the immune system. Using early-life programming via in ovo stimulation with probiotics and prebiotics, it may be possible to avoid selected metabolic disorders, poor immunity, and pathogen resistance, which the broiler industry now faces due to commercial hatching and selection pressures imposed by an increasingly demanding market.

## 1. Introduction

The period of embryonic and immediate post-hatching development is approaches 50% of the productive life of broilers [1,2,3]. Under commercial conditions, birds have access to feed 36–72 h after hatching, leading to delays in the development and maturation of muscle, the immune system, and the gastrointestinal tract (GIT) [3,4,5]. The health of the animal’s intestine is directly related to the animal’s feed and surrounding environment. Unlike mammalian embryos, avian embryos have limited nutrients and energy for growth and development coming from the broiler breeder hen [6]. The term “early-life programming” refers to the way in which environmental factors, including nutrition, alter the course of fetal development, resulting in enduring modifications in the structure and function of biological systems [7,8,9,10].

In poultry, the GIT microbiota plays a vital role in host wellness, improving immune responses and nutrient and energy uptake as well as keeping the digestive system healthy [11,12]. Moreover, the early colonization of the poultry gut by beneficial bacteria enables birds to counter the potential environmental and pathogenic challenges [13,14]. The GIT is an open ecological system colonized in mammals immediately after birth by commensal microbes [15].

Maternally acquired microbes establish colonization through the mother’s vaginal microbiota [15]. Unlike mammals, chicks are believed to hatch with a sterile GIT [16]. In natural hatching, commensal microbiota colonizes the gut of newly hatched chick by contact with the hen and nesting boxes. However, in commercial poultry production, chicks are hatched in an artificial incubator rather than in a henhouse, resulting in a shortage of natural and desirable microbiota sources [16]. Consequently, the first exposure to bacteria can carry a risk of infection with pathogens.

The perinatal period is characterized by rapid developmental changes of the small intestine and immune system [3,17]. Hence, the early-life inoculation of beneficial microbiota has particular importance in chicken life, which directs the bird’s behavior, health, and productivity [9,11,15]. Commensal gut microbiota competes directly with pathogens and mediates the maturation and differentiation of the host’s intestinal epithelium and immune system [18].

Owning a healthy GIT with optimum structure and function is necessary for meat-type chickens with a high growth rate to meet the optimum growth performance [19]. Early initiation of gut colonization with the commensal microbiota is essential in order to develop the immunological defense and jump-start the maturation of the GIT [20,21,22]. Therefore, perinatal inoculation with beneficial bacteria and the non-digestible feed ingredients that promote probiotic bacteria or their combinations may modulate the gut microbiota and mediate the early-life programming in broiler chickens.

A broad use of probiotics, prebiotics, and synbiotics has taken place over the last decade as natural bioactive compounds. Several kinds of water-soluble bioactive compounds have been utilized in animal diets with many benefits [1,23,24,25,26,27,28,29,30]. In ovo technology, through which a tiny quantity of material is injected into the bird’s egg/embryo during incubation, offers a novel alternative for delivering bioactive compounds to the chick embryo before hatching. Recent experiments from various researchers showed the benefits of in ovo feeding technology on body weight, feed conversion ratio, and pectoral meat yield [31,32,33,34].

However, in ovo feeding has an epigenetic capability to modulate the expression of metabolic-related genes and the development of vital tissues and, consequently, affect poultry performance [35,36]. Therefore, early-life programming and even incubation procedures may be revolutionized by this new technology. Supplementing the egg membranes with appropriate nutrients or bioactive compounds (by the in ovo technique) is a novel way to improve embryo health and a means to jump start development.

By definition, a prebiotic is a selectively fermented ingredient that allows specific changes in the composition or activity of the GIT microbiota that confers benefits on the host’s health and well-being [37]. The mode of action for prebiotics includes competitive exclusion that allows for the growth of intestinal microbiota (mainly *Bifidobacteria* spp.) in the GIT and limits pathogens and toxins. Prebiotics stimulate both nonspecific and specific (macrophages and B and T lymphocytes) components of the immune response resulting in improved defense against viral, bacterial, fungal, and parasitic infections [38].

On the other hand, probiotics are defined as live microorganisms that, when administered in adequate amounts, confer a health benefit to the host. Most probiotic microorganisms are Gram-positive lactic acid bacteria, such as *Bacillus* spp., *Lactobacillus* spp., *Bifidobacteria* spp. and *Lactococcus* spp., which modulate the intestinal microbiota through colonization of the GIT and inhibition of the growth of pathogenic bacteria. Probiotics prevent the growth of pathogenic bacteria via certain mechanisms (competitive exclusion, producing bactericidal molecules, producing organic acids, and acidifying the large intestine through nutrient fermentation), thus, preventing them from binding to or penetrating the mucosal surfaces [39].

They also play an essential role in enhancing the barrier function of epithelia and altering immunoregulation (decreasing pro-inflammatory effects and promoting protective molecules) [40]. In ovo feeding technology may provide new insights and future directions of prenatal nutrition and stimulation, affecting the early development and later phenotype that will open opportunities for higher production efficiency and poultry welfare.

This review aims to discuss the development and functions of gut microbiota in the gastrointestinal tract of broiler chickens and the factors that can modulate its composition, with particular focus on the in ovo stimulation of the embryonic microbiome as a novel tool for early-life programming in broiler chickens.

### 1.1. Development of Gut Microbiota in Newly Hatched Chicks

For a long time, scientists assumed that eggs are formed in an internal sterile environment and chicks hatch with a sterile gut. However, new technologies in the field, such as next-generation sequencing techniques, have shed light on the fact that this conventional idea is no longer accurate [41]. These techniques allowed scientists to demonstrate that pathogenic bacteria can transfer vertically or horizontally to eggs and subsequently to the chick’s gut [42,43].

The microbiota diversity was found to be increased throughout the reproductive tract, beginning at the infundibulum and extending to the cloaca, but the communities present in the vagina, uterus, and other reproductive tract regions were clearly distinct. Though some parts of the reproductive tract are resistant to various pathogens due to the secretion of lysozyme and other antimicrobial proteins in these parts [44,45], some pathogenic bacteria are present in their microbiome [42]. Hence, pathogens can contaminate eggs during egg formation and oviposition.

Part of the bacteria may penetrate the egg after laying and transmit to the embryo during embryonic development and could initially establish the first intestinal microbiota of the chick. Akinyemi et al. [41] studied the dynamic changes of gut microbiota and the metabolic pathways in chicken embryos across the different stages of embryonic development, providing an insight into the understanding of dynamic changes of gut microbiota during this critical period [41]. They demonstrated an increase of microbial communities in GIT as the embryo developed from day 3 to day 12 of incubation. However, a reduction in microbial populations was observed on day 19 of incubation, and some microbes had disappeared [41].

A previous study found a moderate association between the intestinal microbiota of the embryos and neonatal chicks [11]. Recently, investigators identified about 21 shared genera in the hen oviduct, eggshell, egg white, and the gut of the embryo [46]. Interestingly, egg white and the embryo’s gut exhibited similar microbial structures. In a more recent study, *Escherichia-Shigella* and *Enterococcus* were two dominant genera in the maternal oviduct and might also be found in yielded eggs [42]. These findings dispute the idea that avian eggs and their embryos are entirely sterile before hatching, as evidenced by the fact that embryos swallow yolk by the latter stages of embryonic development. Following this evidence, it now appears that the assumption that the intestinal microbiota is solely acquired on the day of hatching is incorrect [41,46,47].

Compared to mammals, chickens have a shorter GIT, accelerating the digestive transit time and influencing the diversity of the bird’s gut microbiota [48,49]. Due to this anatomical characteristic, chickens have a significantly distinct gut microbiota compared with other food animals [50,51]. Early exposure to microbes is essential for establishing a healthy gut microbial population. The initial inoculum may have long-term effects on the immune system and the microbiota of the intestine in broiler chickens.

The microorganisms of the surrounding environment gradually inhabit the GIT of newly hatched chicks. Accordingly, adult-type microbiota can colonize the gut of newly hatched chicks at a young age or before hatching when given early in life through the proper routes, leading to a developed immune system and improved resistance to harmful bacterial [52]. Many lines of evidence showed that early life experiences have a significant impact on the subsequent vulnerability to chronic illnesses, indicating the beneficial and the long-term effects of early-life programming in broiler chickens [7,53,54].

Immediately after hatching, the GIT of chicken comes into touch with external microorganisms and provides a warm home for a complex microbiome mainly composed of anaerobic bacteria, predominant in the microbiome. Microbiomes become more varied as their host develops, and eventually they attain a reasonably stable but dynamic state [55]. The perinatal period is characterized by the rapid growth and development of GIT. In addition, the gut microbiota of the neonatal chick also develops rapidly within the first three days post-hatching.

Dominant microbial communities are similar in all intestinal segments within young chicks. This similarity gradually decreases with age, and then dominant microbial communities become specific for each intestinal part. Therefore, every part of the GIT within one bird has its own specific dominant microbial population [56]. Denaturing gradient gel electrophoresis analysis showed that the diversity of dominant bacterial communities increases with age [56].

However, several factors are important in establishing the host-specific microbiota, such as the host genotype, diet composition, immune responses, and surrounding environment. A previous study found that exposure to varied bacterial communities early in life could shape the gut microbial structure and expression of genes controlling the function of ion transport, cell cycle, and chromosome maintenance. These findings indicate that the gut microbiota in neonatal chicks can be manipulated by providing them with the right bacterial makeup in the early stages of life [47].

### 1.2. Function of Gut Microbiota

Modulation of the immune system and gut functions, nutrient exchange, and pathogen exclusion are the primary roles of gut microbiota. The interactions between the gut microbiota and chicken host affect both the innate (such as mucin synthesis and composition) and acquired immune responses [57]. Investigations using germ-free birds revealed that the intestinal microbiota has a significant impact on the immune cells and their release of cytokines, even though the exact mechanism remains unclear [58,59,60].

Establishing an intestinal microbiome may decrease pathogen attachment and colonization in the gut via competitive exclusion (Figure 1), which is thought to be the consequence of several processes, such as binding sites on the gut wall, competition for nutrients, and bioactive metabolites (short-chain fatty acids (SCFAs), bacteriocins, lactate, and hydrogen peroxide) [39]. While the proximal gut digests and absorbs the majority of dietary carbohydrates, bacterial communities in the distal gut ferment and break down the indigestible and remaining digestible carbohydrates [61,62]. There are a wide variety of metabolites produced by the fermentation of carbohydrates and proteins by bacteria, including SCFAs [63]. SCFAs impact the health of the host in two ways: as energy sources and as signaling molecules (Figure 1).

Fiber-related compounds transformed by bacteria are linked to lower rates of many chronic illnesses. A bird’s digestive system (from crop to ceca) shows signs of fermentation. However, the majority of this fermentation occurs in this bacterially packed region of the digestive system ceca [55]. SCFAs have been linked to increased enterocyte development and proliferation (Figure 1), which may partly explain why the gut microbiota has such a stimulatory impact on intestinal development [64].

It was shown that feeding chicken fermentable carbohydrates, which may promote microbial fermentation and subsequently the formation of SCFAs, increased the intestinal weight. In other words, there are strong associations between the physical structure of GIT and the makeup of microbial populations with spatial heterogeneity along the GIT in poultry [65,66].

*Lactobacilli*-dominant microbiota in the chicken’s crop provides a first line defense against pathogens by reducing the passage of harmful bacteria further along the GIT [67,68]. Furthermore, *Lactobacilli* play an essential role in utilizing exogenous enzymes by acidifying the crop environment and its content via pre-gastric fermentation [69,70]. Thus, disturbance of the microbial community in the crop may negatively affect enzyme activity. Like the other segments of the GIT, the development of crop microbiota can be affected by age and diet [71]. However, it is typically stable and can be detected in the crop of newly hatched chicks at one hour post-hatching [72].

Therefore, bioactive substances, such as probiotics, prebiotics, or their combinations, may improve crop function, improving the overall health of the digestive system. The use of *Lactobacillus* probiotics (*L. salivarius* and *L. agilis*) as feed additives (10^10^ colony forming unit (cfu)/kg feed) showed a significant increase in the abundance of dominant bacteria (*Lactobacillus* spp.) and non-dominant bacteria [71]. The early maturity of the crop microbiota after hatching may result in enhancing the resistance against pathogenic colonization. In this regard, introducing probiotics in ovo would appear to be the most suited method of ensuring the earliest possible bacterial colonization.

According to research done on broiler chickens, the diverse bacterial populations inhabit the whole GIT, with the most crucial interactions occurring in the ileum and cecum [66]. However, it is unclear what function the microbiota and SCFAs have in developing the duodenum and jejunum. Gut microbiota and its metabolites in the duodenum and jejunum were thought to evolve together with age and may be linked to forming a healthy gut structure. Therefore, age affects the microbial community’s makeup and their metabolites along the GIT [73].

As a result, improving intestinal health and development in the broilers requires better knowledge of the changing patterns of microbial populations and their metabolites with age, particularly during the post-hatching period. Furthermore, SCFAs play an essential role in energy homeostasis by regulating the lipid and glucose metabolism [74]. SCFAs, particularly acetate, propionate, and butyrate, play a crucial role in regulating the intestinal environment and absorption from the lumen. While the gut epithelium uses butyrate and the liver uses propionate, acetate is the SCFA that is most concentrated in the blood [75,76].

Increasingly, data suggest that acetate has a critical role in regulating inflammation and the defense against pathogen invasion. Cyclin gene expression and proliferation of epithelial cells were shown to be influenced by acetate and lactate in vitro in a pH-dependent manner [77]. Even while butyrate’s function in controlling inflammation, cellular differentiation, and apoptosis has long been established, new research shows that it may also help prevent colorectal cancer in mammals [78]. Moreover, butyrate was recently found to be the most potent AP-1 signaling pathway activator in epithelial cell lines [79].

The first few days after hatching are important for the development and health of poultry because the hatchlings are shifting from yolk-based nutrition to carbohydrate-based diets. The digestive system in newly hatched chicks is changed both anatomically and physiologically, making it one of the most rapidly developing organs during the perinatal period [17,80,81,82].

This allows for a rapid shift from one nutrition source to another. Moreover, the quickly developing GIT is an excellent environment for the colonization of microorganisms. Meanwhile, the gut microbiota is critical for intestinal development, intestinal gene expression, and intestinal wall thickness [47,83]. Supplementation with commercial probiotics and mannan oligosaccharide increased the villus height and villus area in poults [84].

The viscoelastic mucus gel layer covers the luminal surface of the GIT and acts as a protective shield against harsh gastric conditions. The mucus-gel layer is the initial line of defense for pathogens trying to cross the gut mucosa. Mucins can be histologically divided into two main groups: neutral and acidic; the latter includes sulfated and sialylated mucin types. Polymeric mucin glycoprotein secreted by goblet cells forms the mucus gel.

Due to their heterogeneous oligosaccharide chains, these glycoproteins prevent bacteria from adhering to the epithelial cells. However, mucin’s rich-carbohydrate content makes it ideal for certain bacteria to thrive. Thus, the mucus structure is critical for intestinal barrier maintenance, and changes in the makeup of this barrier, as well as its structural features, have long been linked to GIT diseases [80,85,86].

Previous research using germ-free animals showed that the weight and wall thickness of the small intestine and cecum were lower than those of normal animals [87]. The gut microbiota affects mucin gene expression and mucin types. The results of previous studies showed that the absence of gut microbiota reduced the number and density of neutral and acidic goblet cells and increased the sulfated mucin. In addition, mucin 2 mRNA expression was decreased, and sialylated goblet cells were not detected in germ-free birds compared to conventional ones [88].

These findings indicate the association between gut microbiota and the development of mucin synthesis and secretion and the maturation of the small intestine mucosa of chickens. On the other hand, investigations on rodents have also shown differences in the mucosal shape and mucus composition between germ-free and conventionally reared animals due to intestinal microbiota [89,90].

In addition, germ-free birds had fewer and smaller goblet cells with significant differences in acidic mucin composition compared to conventional birds, suggesting less mucus production and less mucin protection in germ-free birds [91]. A newborn gut without a completely functioning adaptive immunity makes it more vulnerable to illness, indicating that the adequate concentrations of these mucin types during early development may play an important role as a dynamic protective barrier against pathogens [92].

## 2. Modulation of Gut Microbiota

The intestinal microbiota is a dynamic and malleable ecosystem. It is intertwined with the animal’s physiology. The gut microbiota’s malleability offers an interesting opportunity to investigate the microbiota’s mechanisms and the possibility of developing treatments.

Several studies indicated that environmental factors, such as age, diet, genotype, intestinal region, and antimicrobial agents play a crucial role in shaping the microbial composition in the animal gut, above and beyond genetic differences in the hosts (Figure 2). Diet is a primary modulator of microbiota structure and plays a vital role in the microbiome composition and function [93]. Short-term dietary changes dramatically affect the microbiota, while a particular long-term dietary pattern may result in certain microbial patterns [94,95,96].

Studies have suggested that intestinal bacterial colonization happens quite early in the chicken’s life and takes numerous days to stabilize. During this time, all ecological microbes, including commensals and pathogens, would have an equal opportunity to occupy GIT [47]. Furthermore, early intervention strategies using commercial probiotics, prebiotics, or their combination (synbiotics) can help modulate the gut microbiota to prevent the colonization of pathogens and enhance the immune system and intestinal development [13,22,97,98].

In ovo stimulation by *Bifidobacterium bifidum* and *Bifidobacterium longum* through in-yolk sac injection on day 17 of incubation improved the growth performance and ileal development of broiler chickens on day 35 of age [14]. Furthermore, in ovo inoculation of *Bifidobacterium* spp. increased the counts of ileal lactic acid bacteria and *Bifidobacterium* spp. compared to the control. Moreover, the total coliform and bacterial counts decreased with in ovo treatments [13].

On the other hand, in ovo inoculation with *Bacillus* spp. significantly reduced the total number of Gram-negative bacteria at the day of hatching and day 7 of age compared to the control group. Proteobacteria phylum showed a significant reduction, while the Firmicutes exhibited a significant increase due to the in ovo injection with probiotics. This effect of *Bacillus* spp. appears to be due to altering microbiota populations and their community structures [21].

The administration of 20 mg oligosaccharides extract of palm kernel cake/egg through in ovo injection modulated the composition of the total cecal bacteria [99]. In ovo inoculation of prebiotics on day 12 of incubation reduced the severity of intestinal lesions and oocyst excretions induced by the natural infection with Eimeria [100]. These results demonstrated that the in ovo administration of probiotics and prebiotics can modulate the gut microbiota at the early stage of life and have a long-term effect on the composition and community structure.

## 3. In Ovo Technique as a Novel Tool for Early-Life Programming

In ovo vaccination against Marek’s disease was successfully demonstrated as a dependable way to protect against infection from virus exposure during development in the early 1980s [101,102]. Due to the success with the in-ovo vaccination, extensive research was done on the injectable form of biological substances, including amino acids, vitamins, minerals, hormones, immunostimulants, probiotics, prebiotics, and other bioactive compounds [16,31,103,104,105,106].

The primary benefit of in ovo technology is administering bioactive compounds at an early stage of embryonic development, resulting in a long-term effect on bird health (for example, immune responses, gut microbiota, resistance to pathogens, and growth performance). Thus, by targeting embryonic GIT and its natural microbiota, the in ovo approach is able to boost the inception of embryonic gut microbiota, with the goal of preparing it to establish the optimum microbiome before hatching [43].

Studies have shown that long-term environmental impacts throughout early development led to the formation of cellular function and physiological responses; these long-term factors appear to be the developmental roots of chronic illness susceptibility [7,15,54]. During the perinatal period, exposure to pathogenic bacteria impairs the development of immune organs, the GIT, and skeletal muscles, reducing the growth performance and immune responses of the newly hatched chicks [15].

The concept of early-life programming postulates that exposure of the embryo to environmental stressors during the perinatal period shapes the development and maturity of the critical organs, resulting in lasting physiological changes after hatching [107,108,109,110]. It appears that the only applicable method to provide direct stimulation to a growing embryo is in ovo technology, which is thought to help reduce stress during the perinatal period and throughout the lifespan [16].

While most of the immune system development in broilers happens early in life, it is essential to have access to the bioactive compounds that can enhance the development and maturation of the immune system. Probiotics and prebiotics have the ability to stimulate the proper development of the immune system [18,111,112]. Moreover, early stimulation of the gut microbiota via supplementation with probiotics, prebiotics, or their combination can enhance the productivity and health of newly hatched chicks [14,113,114].

Probiotics and prebiotics supplied through the in ovo technique had no adverse effects on the hatchability when delivered with the proper doses. Furthermore, in ovo incorporation of these substances improved the development of lymphoid organs and GIT and gut microbiota [21,31,115].

## 4. Basics of In Ovo Technique

Since the in ovo method was introduced, researchers have focused on exploring how to improve the efficiency of this technique via the consideration of in ovo injection procedures according to embryonic age, injection site, and volume of the injected solution [16,32]. There are five potential injection sites via which an in ovo injection may be administered during the different stages of embryonic development: the air sac, the allantois, the amnion, the yolk sac, and the embryo body [16,32,105].

Bioactive substances injected into the air cell can be transferred into the circulatory system via the highly vascularized chorioallantoic membrane. The embryo orally swallows those injected into the amniotic fluid during the late embryonic development before internal piping [97,113,114]. In ovo injection in the air cell is safe and recommended for several bioactive substances. Moreover, the air chamber appears to be the appropriate site for prebiotics, probiotics and their combinations to be injected after day 12 of incubation [113,114].

In ovo injection with these substances before the allantochorion has grown sufficiently will not penetrate through it and thus cannot reach the embryo’s gut. As almost all in ovo research focuses on the supplement being injected, the variables in those studies generally concentrated on nutrients, hormones, immunostimulants, or other bioactive substances, intending to promote growth and stimulate the immune system. The aim of the in ovo injection of prebiotics, probiotics, and synbiotics is to facilitate the early colonization of the embryonic gut with the native microbiota to establish the beneficial microbiome.

It is also necessary to utilize the proper volume of the solution used for in ovo injection. According to McGruder et al., volumes of electrolyte solutions should not be higher than 2000 μL [116], while Zhai et al. encouraged using less than 700 μL of carbohydrate solutions [117]. The commercial vaccination of Marek’s disease typically uses a volume of 50 μL. Previous studies showed that providing 200 μL of dissolved bioactive stimulus was suitable for prebiotics and synbiotics [118,119].

This includes gene expression changes, improvements in embryo development, and other positive effects. The development of microbial proliferation in the embryonic GIT needs to occur at the proper time. The timing of in ovo stimulation of gut microbiome on day 12 of embryonic development was established via some experiments based on evaluating the proliferation rate of *Bifidobacteria* at the day of hatching [119].

## 5. Gut Microbiota and Intestinal Immune Homeostasis

The immune system has two components that are powerful in adapting and responding to highly diverse challenges: innate mechanisms and adaptive responses [58]. This cellular network works together to maintain and repair host tissue function and respond to new environmental challenges, such as microbial infections and environmental challenges [4,57].

To the best of our knowledge, the immune system, particularly the adaptive immune system, has not developed to its full capacity immediately after hatching or birth in all birds and mammals that have thus far been studied, making these animals vulnerable to various pathogens during the perinatal [58]. Documented associations have also been found between the evolution of the immune system and the makeup of the gut microbiota, providing further evidence that the majority of the immune system has developed to have a symbiotic interaction with these varied microbial populations [57].

Intestinal microbiota and the immune system are closely coupled and help to regulate one another. The gut microbiota interacts with the host immune system, modulating its development and the expression of immune-related genes. Commensal bacteria stimulate immune responses via producing microbial metabolites and antimicrobial compounds. Gut microbiota has a serious influence on host immune system development, training, and function [58,120,121].

Since these microorganisms help the host, the immune system has developed mainly to ensure the symbiotic relationship between the host and these rapidly changing microorganisms. The immune system and gut microbiota maintain each other in this partnership resulting in protection against pathogenic bacteria and maintenance of the homeostasis of commensal microbiota [59,120,122,123].

Unlike mammals, maternal antibodies are transferred to offspring through the deposition of antibodies in the egg during egg formation. In poultry, IgY (IgG) is deposited in the yolk, while IgA and IgM are transferred into the egg white [124]. IgG mostly starts to transfer at a low rate from the yolk sac to the embryo via embryonic circulation as early as day 7 of incubation. At the same time, IgA and IgM are transported from the albumen to the yolk (day 12 of incubation) but not into embryonic circulation [125].

The innate and acquired components of the mucosal immune system work together to form a network. The mucosal immune system has two different functions: it defends against pathogens and tolerates the diverse and beneficial commensal microbes in the intestines without overt inflammation [123]. Commensal microbes activate bacterial metabolic pathways to produce substances that help bacteria avoid, alter, or stay alive even when they are being targeted by the host’s innate immune defenses. The related point is that certain microbial-derived chemicals can also help commensal processes, thus, benefiting the host and its microbiota [57].

The integrity of the intestinal mucosal barrier is essential for the immune system’s function. The first element of the mucosal firewall is the microbial barrier, which is composed of microorganisms that reside in or near the upper mucus layer. These beneficial bacteria help to ensure that pathogen invasion does not occur, hence, supporting immune system defense [55,57]. They generate metabolites or components that impact immunological signals and immunological homeostasis. The second component of mucosal immunity is the mucus-covered gut epithelium.

The mucus helps prevent commensal bacteria from direct contact with the epithelial cells. By expressing their characteristic mucus, goblet cells, antimicrobial peptides, epithelial cells, and mucosal IgA produced by dendritic cells are able to form a distinct layer that separates them from commensals and prevents the pathogens from crossing over into the intestinal mucosa. Intestinal IgA modulates bacterial colonization and microbial gene expression by preventing and promoting it while also having fine-tuned control over the microbiota [15,55,57,86].

The success of gut immune homeostasis is tied to how different immune cells (including regulatory T cells (Tregs), Th17 cells, IgA-secreting B cells, and innate immune cells) work together [57,123]. Tregs play an essential role in maintaining gut immune homeostasis and self-tolerance in GIT, as the gut is constantly exposed to potentially inflammatory antigens [126,127]. The gut microbiota of chicken is dominated by Firmicutes and followed by others, including Actinobacteria, Bacteroidetes, and Proteobacteria [128].

One hypothesis is that, in modern broiler production, practices, including diet changes, hatchery processing, transportation, and high stocking densities, may weaken birds’ immunological capabilities and, therefore, make them more vulnerable to gut infections [11]. To enable the development of antibodies and promote the maturation of the cellular components of the intestinal immune system, it is necessary to establish an appropriate gut microbiota [18,112]. The earlier handling the gut microbiota, the better it is for enhancing immune responses, particularly with regards to increasing the level of IgA that is locally secreted.

Specific bacteria in gut microbiota appear to drastically affect the host immunity due to their adhesion to intestinal tissue and/or release of molecules that modulate the immune system [58,129,130]. For instance, host-associated bacteria possess an adhesive capability to interact with the intestinal epithelial cells, and this interaction allows these bacteria to spread widely to epithelial cells, resulting in the activation of Th17 and immunoglobulin responses (IgA), which, in turn, limits their expansion [92,121,123,131].

The gut microbiota, especially Gram-positive bacteria, are able to regulate CD4^+^ Treg induction via the production of SCFAs. It has been found that reducing the population of gut microbiota was associated with the reduction in the population of CD4^+^CD8^−^CD25^+^ and CD4^+^CD8^+^CD25^+^ T cells in the cecal tonsils of chicken exposed to antibiotics, and the supplementation with acetate enhanced the induction of CD4^+^CD8^−^CD25^+^ T cells via a GPR43-independent pathway (acetate receptor) [127].

The authors found that the administration of acetate increased GPR43 expression on CD4^+^CD8^−^CD25^+^ T cells, which may explain the association between acetate uptake and the recovery of CD4^+^CD8^−^CD25^+^ T cells. In mammals, the induction of Tregs was influenced by the levels of SCFAs, including acetate, propionate, and butyrate, which are produced mainly by Firmicutes and Bacteroidetes following fermentation of dietary fibers [132,133].

SCFAs are identified as the important bacterial metabolites in the gut that aid in maintaining the immune system and the anti-inflammatory aspects. Moreover, SCFAs play an essential role in maintaining gut mucus barrier by their signaling on intestinal mucosal lymphoid via G-protein-coupled receptors. Common enteric pathogens, such as *Escherichia coli,* may alter their pathogenicity profile due to exposure to SCFAs. It was reported that in ovo injection with prebiotics (chitooligosaccharide) enhanced the enrichment of *Lactobacillus* sp., including *L. salivarius* and fatty acid biosynthesis. This modulated the expression of immune-related genes in the gut of broiler chickens [134].

The intestine microbiota modifies the gene expression in response to dietary changes, which leads to variations in the amounts of intestinal immune-stimulating bacterial antigens [93]. The gut immune system is able to recognize and respond to changes in the metabolic state of the microbiota by detecting microbial metabolites via pattern recognition receptors (PRRs) [57].

In other words, the gut microbiota uses multiple metabolic pathways to metabolize both diet- and host-derived compounds [135], some of which ultimately affect elements of the gut immunity [136]. For instance, commensal microbiota catabolizes tryptophan into indole, enhancing barrier functions via inducing the expression of several genes involved in intestinal epithelial cell functions [137]. It is also well known that the microbiota ferments and breaks down plant-based fibers into SCFA with various anti-inflammatory properties.

Overall, intestinal homeostasis is achieved when the host immune system and the microbiota work together to keep each other in balance. Immune system stimulation by the native gut microbiota is necessary for the proper development and regulation of the intestinal immune system. Simultaneously, mucosal immune responses prevent undesirable overgrowth, pathogenic translocation, and unnecessary inflammation [57].

A recent study used different kinds of prebiotics and synbiotics for in ovo stimulation in broiler chickens [111]. They concluded that in ovo administration with synbiotic (inulin and *Lactococcus lactis* subsp. *lactis* IBB SL1 was the most effective stimulator to the immune system [111]. The findings of a recent study also demonstrated that prebiotics or synbiotics showed a great induction of the immune-related genes in the spleen and cecal tonsils of broilers.

Synbiotic groups showed the highest mRNA levels compared with the prebiotic groups [18]. In a previous study [114], the authors investigated the long-term transcriptomic responses in the lymphatic organs and GIT tissues of broilers treated in ovo with prebiotics and synbiotics. Of the four bioactive substances used, galactooligosaccharides were the most effective to stimulate gut–microbiome interactions.

The great bifidogenic impact of galactooligosaccharides strongly down-regulated immune-related genes in cecal tonsils. This effect may be due to the ability of galactooligosaccharides injected in ovo to infiltrate the chorioallantoic membrane and enhance the growth of indigenous microbial communities in the gut of embryos. This microbiota promoted the development and maturation of gut-associated lymphoid tissue, which resulted in the improved tolerance of the local immunity.

The effect of in ovo treatment was shown in the inhibition of cellular and humoral immune responses, indicating the potential role of the complement pathway as a regulator of negative regulation of the inflammatory immune responses in adult broilers that were treated in ovo with the galactooligosaccharides [114] (Table 1).

## 6. Gut Microbiota and Growth Performance

The structure of microbial populations is dynamic and is typically influenced by the host genotype, environment, and age [42]. In poultry, little is known about the effects of genotypic variation on early-life microbial colonization and the functionality of the gut, as this knowledge gap exists for both the host and microbial populations. It is currently believed that, along with the functional development of intestinal mucosal tissue, colonization of the gut microbiota in hatchlings depends on a combination of genetic background and functional development of the intestinal mucosal tissue, including early-life programming of the gut functions [42,94].

In recent years, there has been growing interest in the vital link between the gut microbiota and the growth performance of broiler chickens. Dietary supplementation with some bioactive compounds, such as probiotics, enhanced the growth performance of broiler chickens by improving gut health [140,141,142,143]. Furthermore, the early intervention with some bioactive substances may provide early bacterial colonization in the gut of newly hatched chicken and improve the gut microbiota and, consequently, improve gut health and growth performance (Table 2).

In ovo injection with probiotics (2 × 10^8^ CFU/egg of *Bifidobacterium bifidum, B. animalis, B. longum*, or *B. infantis*) was used in a recent study. The authors found that in ovo inoculation with probiotics did not affect the feed intake. However, the feed conversion ratio was significantly influenced. Birds treated with *B. bifidum* had the best FCR compared to control and other groups [22]. Abd El-Moneim et al. [14] investigated the effect of in ovo inoculation with 109 and 107 CFU/egg *B. bifidum*, and 10^9^ and 10^7^ CFU/egg *B. longum*, respectively, on the growth performance of broiler chickens. They found that body weight gain and FCR were significantly improved in all treatment groups compared to the control groups [14].

To examine the effect of in ovo injection of bioactive compounds at day 12 of incubation, Maiorano et al. [144] carried out an experiment using in ovo injection of prebiotics (1.9 mg of raffinose) and synbiotics (1.9 mg of raffinose and 10^3^ CFU *Lactococcus lactis* ssp. *lactis* SL1/egg, and 1.9 mg of raffinose and *Lactococcus lactis* ssp. *cremoris* IBB SC1, a commercially available synbiotic Duolac, Biofaktor, Skierniewice, Poland). They showed that in ovo prebiotics or synbiotics had no impact on the final body weight.

Prebiotics had no adverse effect on the FCR; however, it was significantly impaired with the treatments of commercial symbiotic and raffinose enriched with *Lactococcus lactis* ssp. *cremoris* IBB SC1 [144]. Similarly, inoculation of probiotics (*Bacillus subtilis*, *Enterococcus faecium,* and *Pediococcus acidilactici*) did not significantly impact broilers’ growth performance [145]. Contrary, chicks treated in ovo with *Bacillus* spp. base probiotic (10^7^ CFU/egg) on day 18 of egg embryogenesis showed an increase in the body weight on the day of hatching and day 7 of age compared with untreated chicks [21].

Two kinds of synbiotics (*Lactobacillus salivarius* with galactooligosaccharides and *Lactobacillus plantarum* with raffinose) were in ovo injected into the air cell of meat-type chicken eggs at day 12 of embryogenesis with doses (10^5^ CFU/egg of probiotic, 2 mg/egg of prebiotic) [146]. They found no effects of the treatments on the body weight and feed consumption [146]. In ovo delivery of a commercial prebiotic (extract of beta-glucans, trans-galactooligosaccharides) and raffinose family oligosaccharides significantly improved body weight gain compared to the control group, particularly throughout the first three weeks [97].

Moreover, prebiotics increased the feed intake and the feed conversion ratio. Injection of this prebiotics in ovo followed by the water administration did not reveal a synergistic effect on broiler performance compared to the in ovo injection alone [97]. Hence, these authors proposed that prebiotic administration through in ovo method can substitute the extended and expensive administration of the bioactive substances post-hatching through drinking water [97].

## 7. Gut Microbiota and Immune Responses

The development and maturation of the immune system, particularly during the early stages of development, is affected by environmental factors, which are collectively known as the microenvironment. Among these factors, the interaction with the healthy microbiome is the most important component of developing the immune system [133]. Immune system can be modulated by probiotics or prebiotics or their combinations. Probiotic bacteria have a beneficial effect on the health of the host and gaining more and more interest as an alternative for antibiotic.

In ovo injection of probiotic, prebiotic, or synbiotic affects the colonization and development of the central and peripheral immunity in broiler chickens [113,130]. This effect may be attributed to the metabolites produced by commensal bacteria [127,133]. In ovo injection of prebiotics (inulin and Bi^2^tos) and synbiotics (inulin + *Lactococcus lactis* subsp. and Bi^2^tos + *L. lactis* subsp.) affected the innate immunity of broiler chickens [112,131]. In this study, authors found that in ovo administration with these compounds can temporarily modify the development, maturation of leukocytes, and their reactivity [112,131].

Previous research has established that in ovo injection of a prebiotic or synbiotic to embryos downregulated mRNA levels of the immune-related genes in the lymphoid organs of broiler chickens (cecal tonsils and spleen) [152]. While synbiotic in ovo upregulated gene expression for IL-4, IL-6, IFN-β, and IL-18 in the spleen. However, the expression of IL-8 was not affected in birds subjected to synbiotic in ovo. Moreover, samples of cecal tonsils showed downregulation for expression of all cytokines, except for IL-18 [121].

The development of spleen and bursa of Fabricius showed significant changes by in ovo treatments. In ovo injection with synbiotic on day 12 of embryonic development increased thymocytes in the thymus cortex of the adult chickens. Moreover, the bursa and bursa to spleen index were higher in birds administered in ovo with raffinose and (*Lactococcus lactis* subsp. *cremoris* IBB SC1 + raffinose) [120]. However, intra-amnion delivered probiotics had no significant effect on antibody titres against Newcastle disease virus [145].

A recent study used different kinds of prebiotics and synbiotics for in ovo stimulation in broiler chickens [111]. They concluded that in ovo administration with synbiotic (inulin and *Lactococcus lactis* subsp. *lactis* IBB SL1 was the most effective stimulator to the immune system [111]. The findings of a recent study demonstrated also that prebiotics or synbiotics showed a great induction of the immune-related genes in the spleen and cecal tonsils of broilers. Synbiotic groups showed the highest mRNA levels compared with the prebiotic groups [18].

## 8. Gut Microbiota and Cecal SCFA Concentration

Research demonstrated that the gut microbiota performed a wide range of physiological and pathological activities in the animal body [66]. The balance of gut microbiota helps maintain intestinal epithelial homeostasis, immune system development, nutrient metabolism, and protection from infections. Figure 3 illustrates the impact of early colonization with beneficial bacteria on the gut microbial balance.

A healthy gut will likely prevent invading pathogens using their own natural resistance mechanisms, they may also defend themselves by secreting antimicrobial chemicals, volatile fatty acids, and chemically modified bile acids [153,154,155]. The composition and metabolic activity of the intestinal microbiota can be influenced by the diet. For instance, probiotic has the ability to decrease the likelihood of infection and affect the metabolic activity of the host by modulating the host gut microbiota [39,65].

Furthermore, probiotics assist in synthesizing vitamins and cytokines, as well as inhibiting cancer development [39]. To be classified as a probiotic, a particular strain must have been scientifically shown to be safe and effective in the context of these characteristics. Many of the physiological effects associated with the fermentable fiber consumption are directly linked to its selective promotion of specific strains of gut microorganisms, particularly bifidobacteria and lactobacilli. In ovo injection with different kinds and doses of oligosaccharides at day 12 of incubation affected hatchability percentage and increased bifidobacteria in the gut of two-day-old chickens [156].

Moreover, another study confirmed that in ovo injection with different doses of raffinose had a long-term effect on the maintenance of a high abundance of bifidobacteria in the adult chickens [157]. The use of different kinds of probiotics (2 × 10^8^ CFU/egg of *Bifidobacterium bifidum*, *B. animalis*, *B. longum*, or *B. infantis*) for in ovo stimulation at day 17 of incubation showed a significant improvement in the count of ileal lactic acid bacteria and *Bifidobacterium* spp. compared to control group, whereas the total coliform and total bacterial counts were significantly decreased [22].

Dunislawska et al. [146] used two forms of synbiotics (*Lactobacillus salivarius* with galactooligosaccarides and *Lactobacillus plantarum* with raffinose) for in ovo injection into the air cell of meat-type chicken eggs at day 12 of incubation at doses (10^5^ cfu/egg of probiotic, 2 mg/egg of prebiotic). They found that synbiotics in ovo administration increased the microbial communities of *Lactobacillus* spp. and *Enterococcus* spp. in the ileum compared to the control group, whereas the bacterial count showed the highest value in the cecum of untreated chicks [146].

Commensal bacteria produce SCFA and lactic acid and a range of other antimicrobial compounds. Additionally, the use of the fermentable fiber (prebiotic) offers a promising approach to restore microbial communities and to support barrier function of the gut epithelia by their prebiotic action. Prebiotics and probiotic microorganisms used together help produce synbiotics, which provide advantages superior to those of prebiotics or probiotics on their own.

In ovo delivery of some commercial prebiotic (extract of beta-glucans and trans-galactooligosaccharides) significantly increased *Bifidobacteriaceae* and *Lactobacillus* count in feces of broiler chicks on the day of hatching [97]. Pacifici et al. [149] demonstrated that stachyose and raffinose enhanced gut health through improving the growth of beneficial bacteria and preventing the abundance of potentially pathogens. Enhancing the abundance of the beneficial bacteria leads to increasing the synthesis of short-chain fatty acid, along with a corresponding increase in Fe solubility.

This may improve Fe absorption by improving the production of short-chain fatty acids by bifidobacteria and lactobacilli, and thus reduce the levels of potentially pathogenic bacteria that use dietary Fe in the colon. Inulin, a prebiotic, has been shown to stimulate the growth of healthy intestinal microorganisms when used as a dietary supplement [158]. Due to the fermentation activity of these beneficial bacteria, *Bifidobacterium* and *Lactobacillus* are able to break down prebiotics, which increases their growth and results in higher SCFA synthesis.

The findings of this study observed that the administration of prebiotics through the in ovo route significantly improved the relative abundance of both *Bifidobacterium* and *Lactobacillus*, whereas the relative abundance of *clostridium* was significantly decreased in the cecal content of the broilers treated with prebiotics [149]. The findings of a previous study showed that acetic acid is the major volatile fatty acid found in broiler feces followed by butyric and propionic acid [151].

Broilers supplied with 0.3% *L. plantarum* + 1.0% inulin were significantly affected by the treatment and had the highest level of acetic acid and total volatile fatty acids. The highest propionic acid was found in birds that received 0.3% *L. plantarum* + 0.8% inulin. However, the concentration of butyric acid was not significantly influenced by the treatments [151].

## 9. Conclusions

This review focused on the development and function of the gut microbiota and the relatively new area of in ovo use in the poultry industry that has recently been investigated. A growing body of research indicates that in ovo administration of bioactive compounds as an alternative to antibiotic growth promoters (AGP) has promise for improving avian health and performance.

Early establishment of the gut microbiome via in ovo injection with prebiotics and probiotic strains may have a positive impact on the development of the immune system, intestinal epithelium, and well-being of the chicken host. Next-generation sequencing techniques allowed scientists to demonstrate that pathogenic bacteria can transfer vertically or horizontally to eggs and subsequently to the chick’s gut. Under these conditions, in ovo stimulation with prebiotics and probiotics may control and prevent the early exposure to infections when the chicks are most susceptible immediately post-hatching.

## Figures and Tables

**Figure 1 animals-11-03491-f001:**
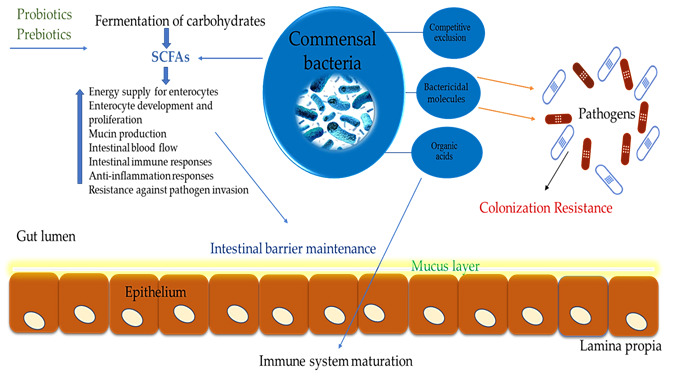
Potential mechanisms by which probiotics and prebiotics contribute to gut health. The microbial fermentation of non-digestive carbohydrates yields SCFAs and improves gut function and overall health. SCFAs serve as an energy source for enterocytes, leading to the maintenance of intestinal integrity. Probiotics and prebiotics modulate gut microbiota via different mechanisms, including competitive exclusion and the synthesis of bactericidal molecules, thus, preventing the colonization of pathogens and enhancing the maturation of the immune system.

**Figure 2 animals-11-03491-f002:**
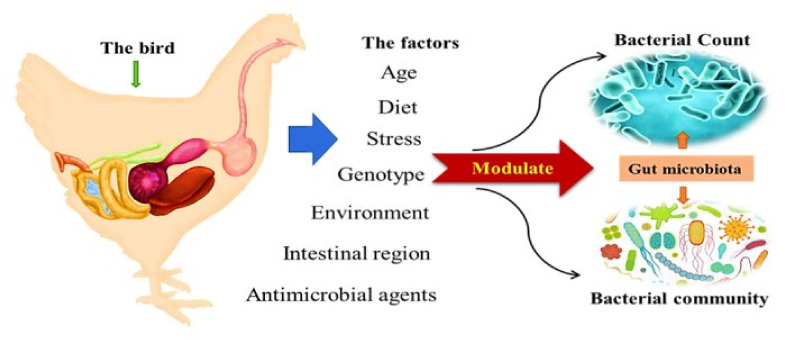
Factors affecting gut microbiota.

**Figure 3 animals-11-03491-f003:**
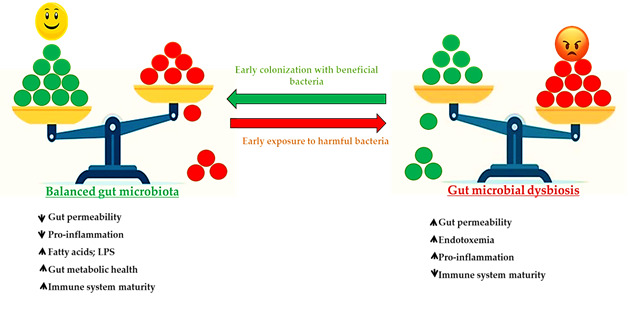
Effect of early colonization with beneficial bacteria on gut microbial balance (red color reveal to pathogens and green color indicate the beneficial bacteria).

**Table 1 animals-11-03491-t001:** Effect of probiotics, prebiotics, and synbiotics injected in ovo on immune responses.

Bioactive Compound	Description and Dose	Site and Time of Injection	Findings	Ref.
Synbiotic (S)	S1. (Lactobacillus salivarius 10^5^ cfu + 2 mg galactooligosaccharides (GOS))/egg S2. (Lactobacillus plantarum 10^5^ cfu + 2 mg raffinose)/egg	Amnion on day 18 of incubation	S1 activated mostly genes involved in immune processes	[138]
Prebiotic and synbiotic	Probiotic (1.76 mg inulin/egg). Synbiotic (1.76 mg of inulin + 1000 CFU *Lactobacillus lactis* subsp. *lactis* IBB2955)/egg	Air cell on day 12 of incubation	Prebiotic or synbiotic had a powerful effect on gene expression in the spleen and cecal tonsils of broiler chickens. The effect of synbiotic was greater than those of the prebiotic.	[18]
Prebiotic	Oligosaccharides extracted from palm kernel cake (20 mg/egg).	Air cell on day 12 of incubation	In ovo injection of prebiotics increased IgG production and antioxidant capacity in serum and liver of prenatal chicks.	
Prebiotics and synbiotics	Inulin (1.76 mg), trans-galactooligosaccharides (0.528 mg), (1.76 mg inulin + 1000 cfu *L. lactis* ssp. *lactis*), and (0.528 mg trans-galactooligosaccharides + 1000 cfu *L. lactis* ssp. *cremoris*)	Air cell on day 12 of incubation	The authors concluded that in ovo administration with synbiotic (inulin and *Lactococcus lactis* subsp. *lactis* was the most effective stimulator to the immune system	[111]
Prebiotics	Galactooligosaccharides (3.5 mg/egg)	Air cell on day 12 of incubation	Galactooligosaccharides administered in ovo down-regulated the expression of immune-related genes that were activated by heat stress.	[139]
Probiotics	2 × 10^8^ cfu of *Bifidobacterium bifidum*, *B. animalis*, *B. longum*, or *B. infantis*	Yolk sac, on day 17 of incubation	The in ovo injection of *Bifidobacterium* improved the immune responses of broiler chickens and increased immunoglobulin levels (IgG, IgM, IgA, and total Igs) in the serum of the broilers.	[13]

**Table 2 animals-11-03491-t002:** Effect of probiotics, prebiotics, and synbiotics injected in ovo on gut health and growth performance.

Bioactive Compound	Description and Dose	Site and Time of Injection	Findings	Ref.
Probiotics alone or in combination	1. Marek’s Disease (HVT) vaccination as a control group. 2. *L. animalis* (∼10^6^ cfu/50 μL). 3. *E. faecium* (∼10^6^ cfu/50 μL). 4. *L. animalis* + *E. faecium* (∼10^6^ cfu & ∼10^6^ cfu/50 μL each).	Amnion on day 18 of incubation	The length, weight, and pH of gastrointestinal tissue were affected by in ovo probiotic, resulting in increased FCR from days 7 to 14.	[147]
Synbiotic (S)	S1. (Lactobacillus salivarius 10^5^ cfu + 2 mg galactooligosaccharides (GOS))/egg S2. (Lactobacillus plantarum 10^5^ cfu + 2 mg raffinose)/egg	Amnion on day 18 of incubation	S2 up-regulated expression of genes involved in metabolic pathways	[138]
Prebiotic	Extract of Laminaria species of seaweed 0.88 mg/egg.	Air cell on day 12 of incubation	On day 42 of age, there was no significant effect of prebiotic injection on the growth performance of broiler chickens. In ovo treatment showed a significant increase in villi width and crypt depth on d 21 of age. Prebiotics injected in ovo impaired villus height, width, and surface area in the duodenum compared to the control group.	[148]
Prebiotics	Stachyose (1. 5% and 2. 10%/mL) Raffinose (3. 5% and 4. 10%/mL)	Amnion on day 17 of incubation	There was a significant increase in the relative expression of brush border membrane functioning proteins and villus surface area, as well as a reduction in the relative expression of Fe-related proteins in birds treated with probiotics. Probiotics significantly lowered the relative abundance of harmful bacteria while enhanced that of probiotics. Fe bioavailability, brush border membrane function, and gut microbiota were all positively influenced.	[149]
Prebiotic	Trans-galactooligosaccharides 3.5 mg/egg	Air cell on day 12 of incubation	Prebiotics improved growth performance and carcass weight of chickens at six weeks of age. Prebiotics reduced severity of intestinal lesions and oocyst excretion induced by natural infection with *Eimeria*.	[100]
Prebiotics and synbiotics	Inulin (1.76 mg), trans-galactooligosaccharides (0.528 mg), (1.76 mg inulin + 1000 cfu *L. lactis* ssp. *lactis*), and (0.528 mg trans-galactooligosaccharides + 1000 cfu *L. lactis* ssp. *cremoris*)	Air cell on day 12 of incubation	No significant effects of probiotics and synbiotics were observed on FCR. However, trans-galactooligosaccharides and inulin + *Ls lactis* subsp. *lactis* significantly increased final body weight of treated chickens.	[150]
Probiotic	*Bacillus subtilis* fermentation extract 10 × 10^6^ cfu/egg	Amnion on day 18.5 of incubation	In ovo administration of the probiotic improved intestinal morphology without impairing hatch performance or gut homeostasis.	[151]
Probiotic	*Bacillus* spp. base probiotic 10^7^ cfu	Amnion on day 18 of incubation	Probiotics administered in ovo decreased the severity of virulent E. coli horizontal transmission and infection in broiler chickens during the hatching period.	[21]

## Data Availability

Not applicable.

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
