# Peer review of "Managing Gut Microbiota through In Ovo Nutrition Influences Early-Life Programming in Broiler Chickens"

_animals, 2021, doi:10.3390/ani11123491_

Round 1
Reviewer 1 Report
The manuscript provided a review on "Managing Gut Microbiota through in ovo Nutrition Influences Early-Life Programing in Broiler Chickens. I think that the paper is well prepared. However there are some minor grammatical errors that should be addressed.
Comment 1: Ln 42. Delete "affecting"
Comment 2: Ln 74: Reference provided in different format.
Comment 3: Ln 108: correct to ......"and a means to jump start development".
Comment 4: Ln 136: correct to read...."eggs are formed in an internal...."
Comment 5: Ln 150: The reference is provided in different format that the rest.
Comment 6: Ln 156: Replace "were" with "Had"
Comment 7: Ln 195: Specify which category/class of genes were influenced by the bacterial community in the referenced paper.
Comment 8: Ln 258: replace "newborn chicks" with "newly hatched chicks".
Comment 9: Ln 269: delete "of which".
Comment 10: Ln 329: Reference format is different from the rest.
Comment 11: Ln 408-410: Rewrite this segment its not very clear.
Comment 12: Ln 539: delete "growing recognition of"
Comment 13: Ln 570: delete " have been found"
Comment 14: Change font on figure 3 to make it more legible
Author Response
Academic Editor Comments: Dear authors, although I decided to continue to peer-review the manuscript, I suggest you to look into Figure 3, readability is poor and should be amended
Response
Thanks a lot for this comment, we improved the figure as suggested.
Reviewer 1#
The manuscript provided a review on "Managing Gut Microbiota through in ovo Nutrition Influences Early-Life Programing in Broiler Chickens. I think that the paper is well prepared. However there are some minor grammatical errors that should be addressed.
Comment 1: Ln 42. Delete "affecting"
Response
Thanks for this comment, done as suggested.
Comment 2: Ln 74: Reference provided in different format.
Response
Thanks for this comment, done as suggested Ln 77.
Comment 3: Ln 108: correct to ......"and a means to jump start development".
Response
Thanks for this comment, done as suggested Ln 111, 112.
Comment 4: Ln 136: correct to read...."eggs are formed in an internal...."
Response
Thanks for this comment, done as suggested Ln 139.
Comment 5: Ln 150: The reference is provided in different format that the rest.
Response
Thanks for this comment, done as suggested Ln 153.
Comment 6: Ln 156: Replace "were" with "Had"
Response
Thanks for this comment, done as suggested Ln 159
Comment 7: Ln 195: Specify which category/class of genes were influenced by the bacterial community in the referenced paper.
Response
Thanks for this comment, done as suggested Ln 198, 199.
Comment 8: Ln 258: replace "newborn chicks" with "newly hatched chicks".
Response
Thanks for this comment, done as suggested Ln 279.
Comment 9: Ln 269: delete "of which".
Response
Thanks for this comment, done as suggested.
Comment 10: Ln 329: Reference format is different from the rest.
Response
Thanks for this comment, done as suggested Ln 350.
Comment 11: Ln 408-410: Rewrite this segment its not very clear.
Response
Thanks for this observation, we rewrote it as suggested Ln 430-433.
Comment 12: Ln 539: delete "growing recognition of"
Response
Thanks for this comment, we deleted it as suggested.
Comment 13: Ln 570: delete " have been found"
Response
Thanks for this comment, we deleted it as suggested
Comment 14: Change font on figure 3 to make it more legible
Response
Thanks for this observation, done as suggested.
Reviewer 2 Report
The manuscript provided an overview on the effects of in ovo feeding on gut development during the early life in broiler chickens. This review article expanded our knowledge on the early nutrition programming and well-summarized the recent findings. Please, find my comments below
- I would suggest to add a couple of tables to summarized the recent findings
- In Fig. 3 adjust the font to be easily read.
Author Response
The manuscript provided an overview on the effects of in ovo feeding on gut development during the early life in broiler chickens. This review article expanded our knowledge on the early nutrition programming and well-summarized the recent findings. Please, find my comments below
I would suggest to add a couple of tables to summarized the recent findings
Response
Thank you for your review of our paper. We have added two tables as suggested.
In Fig. 3 adjust the font to be easily read.
Response
We improved the figure as suggested.
Reviewer 3 Report
Dear Corresponding Author,
Please, find comments and suggestions listed line-by-line below. Next, clearly and in detail respond to all reviewer's queries and doubts.
Title - the Reviewer does not understand the term „early-life programming”, it should be more specific/detailed.
L8 - vkpaswan.vet@gmail.com - please, change to organization mail.
L14 - aeabdelmoneim@gmail.com - as above, please, do not use the private mail addresses.
L23 - redaelmazoudy@yahoo.com - as above
L25 - dr.mayadarf@gmail.com - as above
L28 - dr.mahmoud.alagwany@gmail.com - as above
L29 - please correct edit errors.
Simple summary
L36 - ‚selected’ instead of „some of”
Abstract
L38 - „is home” please, do not use a colloquial speech
L48-50 - it is not true due to the fact that there is possible to apply the special system of feeding and watering birds directly after hatching. In various companies the name of technology differs,s however, in this case, it is a „HatchCare”, nevertheless, the results are the same. See the example below,
https://www.youtube.com/watch?time_continue=38&v=yIzEeGM_2QU&feature=emb_logo
L51 - microbiota instead of microflora, please, change the whole manuscript in this case.
L56 - from the reviewer's point of view, first the Authors should explain the „early-life programming” after that the term can be used in the manuscript. The best place for this will be „introduction”. Please, di not use this term in the abstract section.
L57 - selected instead of „some of”
Comment 1 - the abstract section is definitely too long. Please, find the information about the number of words in the Guidance for Authors.
Comment 2 - What is a novelty of the current „review paper”? In the available literature, there is plenty of already published scientific articles about the in ovo modulation of the GIT microbiota, physiology, etc. for instance,
1) Rubio, L. A. (2019). Possibilities of early life programming in broiler chickens via intestinal microbiota modulation. Poultry Science, 98(2), 695-706.
2) Rojman Khomayezi & Deborah Adewole (2021) Probiotics, prebiotics, and synbiotics: an overview of their delivery routes and effects on growth and health of broiler chickens, World's Poultry Science Journal, DOI: 10.1080/00439339.2022.1988804
3) Oladokun, S., & Adewole, D. I. (2020). In ovo delivery of bioactive substances: an alternative to the use of antibiotic growth promoters in poultry production—a review. Journal of Applied Poultry Research.
4) Leão, A. P. A., Alvarenga, R. R., & Zangeronimo, M. G. (2021). In ovo inoculation of probiotics for broiler chickens: Systematic review and meta-analysis. Animal Feed Science and Technology, 115080.
Introduction
L70 - the GIT microbiota instead of the gastrointestinal microbiota
L74 - „(Abd El-Moneim et al. 2020; Abdel-Moneim et al. 2020) „ please, find the information about the citation editing requirements. Furthermore, add the full stop at the end of the sentence.
L77- „Unlike mammals, chicks were believed to hatch with a sterile GIT [13].” It is not true. It is well-documented that the GIT of the birds after they hatch is not sterile. Please, find an example below,
Meijerink, N., Kers, J. G., Velkers, F. C., van Haarlem, D. A., Lamot, D. M., de Oliveira, J. E., ... & Jansen, C. A. (2020). Early life inoculation with adult-derived microbiota accelerates maturation of intestinal microbiota and enhances NK cell activation in broiler chickens. Frontiers in veterinary science, 7, 965.
L78,81,85,91,..etc… - microbiota instead of microflora
L77-82 - from the reviewer's point of view the paragraph should be remodeled, due to the fact that the GIT microbiota populations of the birds at hatch are mainly based on the Enterococcus, Escherichia, and Clostridium. The Lactobacillus and other probiotic bacteria are in the minority. However, in general, the birds survive these conditions due to particularly the innate immune defense.
S. Lim, S. Cho, K. Caetano-Anolles, S.G. Jeong, M.H. Oh, B.Y. Park, H.J. Kim, S. Cho, S.H. Choi, S. Ryu
Developmental dynamic analysis of the excreted microbiome of chickens using next-generation sequencing
J. Mol. Microbiol. Biotechnol., 25 (2015), pp. 262-268
L86 - please, add the full stop
L93 - feed instead of food; probiotic instead of „healthy”
L105 - „later on” - please, do not use the colloquial speech.
L114 - in the case of chicken the changes in the large intestine are too far… so the prebiotics should positively affect also small intestinal microbiota… however, the crop and gizzard microbiota may affect the bacteria populations in the hindgut. Thus, it is beneficial to affect the microbiota as soon as possible in the bird's gastrointestinal tract.
L124 - „acidifying the colon”, the colon only?
L172 - also the term „flora” is incorrect, please change to microbiota
L179 - „early life-life programming” please, correct.
Comment 3 - from the reviewer's point of view the quality of figure 1 is poor. The suggestion is to remove this figure or improve the quality, as well as the content substantially. Comparable to other already published figures the presented one is unacceptable.
L224 - „from crop to cecum” to ceca
L226 - „i.e., ceca” instead of „called the cecum”
L234 - „gastrointestinal” - GIT
L236 - „the most crucial interactions” in the reviewer's point of view one of the most important part of the GIT of birds is the crop which may affect the population in the further parts. However, the Authors omitted/avoided this GIT segment in the manuscript.
KieroÅ„czyk, B., Rawski, M., DÅ‚ugosz, J., ÅšwiÄ…tkiewicz, S., & Józefiak, D. (2016). Avian crop function–a review. Ann. Anim. Sci, 16(3), 653-678.
Classen, H. L., Apajalahti, J., Svihus, B., & Choct, M. (2016). The role of the crop in poultry production. World's Poultry Science Journal, 72(3), 459-472.
L261,275 - „gastrointestinal system” GIT
L276-278 - „Previous research using germ-free hens showed that the weight and wall thickness of the small intestine and cecum were lower than those of normal chickens.” Please, add citations.
L302 - „driver” please, do not use a colloquial speech
Figure 2 - this figure does not add any novel information; thus, from the reviewer's point of view should be removed.
L346 - „native microbiota” it means commensal? Please, change
L441 - „firewall” please, do not use a colloquial speech
L476, 478, 480l481- in terms of „CD4+CD8−CD25+ and CD4+CD8+CD25+” please, use the upper index as in the original paper [112].
L550 - there is a necessity to add the upper indexes.
L552- „significantly improved b in all treatment groups” please, correct.
L555 - „Maiorano and his colleagues” -please, change as follow, Maiorano et al.
L563 - „didn't significantly impact” please, check the grammar
L592 - „Bi2tos” is it correct?
Figure 3 - the quality of the figure is absolutely poor, please, correct.
L649 - „Bifidobacterium” please, use italics
651, 666 - „and colleagues” - please, change to „et al.”
658 - „short-chain fatty acids (SCFA)” change to „SCFA”
665 - „Bifidobacteriaceae and Lactobacillae” facility without the italics
L672 - potentially pathogenic bacteria
L679 - „Clostridium” Please use italics
L680-685 - the sentence have to be removed due to the fact that the numerical changes cannot be taken into consideration (the result is non-significant).
L687, 690 - L. plantarum - please, use italics; inulin
Author Response
Dear Corresponding Author,
Please, find comments and suggestions listed line-by-line below. Next, clearly and in detail respond to all reviewer's queries and doubts.
Thank you for your review of our paper. We have answered your point below.
Title - the Reviewer does not understand the term „early-life programming”, it should be more specific/detailed.
Response
The term "early-life programming" refers to the way in which environmental factors, including nutrition, alter the course of fetal development, resulting in enduring modifications in the structure and function of biological systems. Recently, the concept of early-life programming has been well-introduced. Please review the following articles [1-4]. We explained this term in the section of Introduction.
L8 - vkpaswan.vet@gmail.com - please, change to organization mail.
Response
In this time, we have some changes in email addresses in our Institute. This email is currently available for communication.
L14 - aeabdelmoneim@gmail.com - as above, please, do not use the private mail addresses.
Response
In this time, we have some changes in email addresses in our Institute. This email is currently available for communication.
L23 - redaelmazoudy@yahoo.com - as above
Response
In this time, we have some changes in email addresses in our Institute. This email is currently available for communication.
L25 - dr.mayadarf@gmail.com - as above
Response
In this time, we have some changes in email addresses in our Institute. This email is currently available for communication.
L28 - dr.mahmoud.alagwany@gmail.com - as above
Response
In this time, we have some changes in email addresses in our Institute. This email is currently available for communication.
L29 - please correct edit errors.
Response
Done as suggested.
Simple summary
L36 - ‚selected’ instead of „some of”
Response
Done as suggested Ln 36.
Abstract
L38 - „is home” please, do not use a colloquial speech
Response
Corrected as suggested Ln 38.
L48-50 - it is not true due to the fact that there is possible to apply the special system of feeding and watering birds directly after hatching. In various companies the name of technology differs,s however, in this case, it is a „HatchCare”, nevertheless, the results are the same. See the example below,
https://www.youtube.com/watch?time_continue=38&v=yIzEeGM_2QU&feature=emb_logo
Response
We agree with the reviewer's concern. However, the application of early feeding and watering systems is relatively new and limited to certain companies and regions around the world. Therefore, the problem of late feeding still exists and needs effective solutions in many countries of the world. Nowadays, more than 65 billion meat-chickens are produced annually all over the world by highly specialised industries. Only, less than 2% of this amount may be produced according to the method that reviewer has mentioned above, particularly in Netherlands, and the majority are still produced commercially by the traditional practice.
L51 - microbiota instead of microflora, please, change the whole manuscript in this case.
Response
Done as suggested.
L56 - from the reviewer's point of view, first the Authors should explain the „early-life programming” after that the term can be used in the manuscript. The best place for this will be „introduction”. Please, di not use this term in the abstract section.
Response
The concept of early-life programming is widely known. Therefore, it is often used in scientific research as it is without explanation, whether in the abstract or other parts of the manuscripts. However we explained it in the section of introduction. Please, see the below references:
- https://doi.org/10.1038/s41467-020-19638-4
- https://doi.org/10.1016/j.biopsycho.2010.01.007
- https://doi.org/10.1186/1741-7015-12-33
L57 - selected instead of „some of”
Response
Done as suggested Ln 57.
Comment 1 - the abstract section is definitely too long. Please, find the information about the number of words in the Guidance for Authors.
Response
The words number in the abstract is only 290 words!!
Comment 2 - What is a novelty of the current „review paper”? In the available literature, there is plenty of already published scientific articles about the in ovo modulation of the GIT microbiota, physiology, etc. for instance,
1) Rubio, L. A. (2019). Possibilities of early life programming in broiler chickens via intestinal microbiota modulation. Poultry Science, 98(2), 695-706.
2) RojmanKhomayezi& Deborah Adewole (2021) Probiotics, prebiotics, and synbiotics: an overview of their delivery routes and effects on growth and health of broiler chickens, World's Poultry Science Journal, DOI: 10.1080/00439339.2022.1988804
3) Oladokun, S., &Adewole, D. I. (2020). In ovo delivery of bioactive substances: an alternative to the use of antibiotic growth promoters in poultry production—a review. Journal of Applied Poultry Research.
4) Leão, A. P. A., Alvarenga, R. R., &Zangeronimo, M. G. (2021). In ovo inoculation of probiotics for broiler chickens: Systematic review and meta-analysis. Animal Feed Science and Technology, 115080.
Response
Unlike previous works, in this manuscript, we discussed the role of the gut microbiota, mechanisms of action, and its development in newly hatched chicks in some detail that differs from what was previously published with an updated overview. Also, we discussed the methods of manipulating the gut microbiota in light of the technology of in ovo nutrition using prebiotics, probiotics and synbiotics in great detail, unlike the previous works that discussed in general.
In this study, we also discussed in clear details the effect of using these bioactive substances on improving growth, intestinal health, and immunity, especially on the level of gene expression. Overall, to the best of our knowledge, no previous study has discussed this topic in this details that we have updated here.
Introduction
L70 - the GIT microbiota instead of the gastrointestinal microbiota
Response
Done as suggested.
L74 - „(Abd El-Moneim et al. 2020; Abdel-Moneim et al. 2020) „ please, find the information about the citation editing requirements. Furthermore, add the full stop at the end of the sentence.
Response
Done as suggested Ln 77.
L77- „Unlike mammals, chicks were believed to hatch with a sterile GIT [13].” It is not true. It is well-documented that the GIT of the birds after they hatch is not sterile. Please, find an example below,
Meijerink, N., Kers, J. G., Velkers, F. C., van Haarlem, D. A., Lamot, D. M., de Oliveira, J. E., ... & Jansen, C. A. (2020). Early life inoculation with adult-derived microbiota accelerates maturation of intestinal microbiota and enhances NK cell activation in broiler chickens. Frontiers in veterinary science, 7, 965.
Response
We and the reviewer have the same point of view, but it's obviously that our sentence was misunderstood. We reported that until hatch the chicks were believed to have a sterile GIT. The reviewer said "the GIT of the birds after they hatch is not sterile"; the same meaning! Even the article cited by the reviewer mentioned the same meaning as they said "However, due to hatching in a hatchery environment, colonization in commercial chickens starts with microbiota from environmental, rather than parental sources".
L78,81,85,91,..etc… - microbiota instead of microflora
Response
Done as suggested.
L77-82 - from the reviewer's point of view the paragraph should be remodeled, due to the fact that the GIT microbiota populations of the birds at hatch are mainly based on the Enterococcus, Escherichia, and Clostridium. The Lactobacillus and other probiotic bacteria are in the minority. However, in general, the birds survive these conditions due to particularly the innate immune defense.
S. Lim, S. Cho, K. Caetano-Anolles, S.G. Jeong, M.H. Oh, B.Y. Park, H.J. Kim, S. Cho, S.H. Choi, S. Ryu Developmental dynamic analysis of the excreted microbiome of chickens using next-generation sequencing J. Mol. Microbiol. Biotechnol., 25 (2015), pp. 262-268
Response
We cannot see any contradictions between what the reviewer stated and what we have mentioned in this paragraph.
L86 - please, add the full stop
Response
Done as suggested.
L93 - feed instead of food; probiotic instead of „healthy”
Response
Done as suggested Ln 96.
L105 - „later on” - please, do not use the colloquial speech.
Response
Corrected as suggested Ln 108.
L114 - in the case of chicken the changes in the large intestine are too far… so the prebiotics should positively affect also small intestinal microbiota… however, the crop and gizzard microbiota may affect the bacteria populations in the hindgut. Thus, it is beneficial to affect the microbiota as soon as possible in the bird's gastrointestinal tract.
Response
Corrected as suggested.
L124 - „acidifying the colon”, the colon only?
Response
Corrected as suggested Ln 127.
L172 - also the term „flora” is incorrect, please change to microbiota
Response
Corrected as suggested Ln 175.
L179 - „early life-life programming” please, correct.
Response
Corrected as suggested Ln 182.
Comment 3 - from the reviewer's point of view the quality of figure 1 is poor. The suggestion is to remove this figure or improve the quality, as well as the content substantially. Comparable to other already published figures the presented one is unacceptable.
Response
The quality of Figure 1 has been improved.
L224 - „from crop to cecum” to ceca
Response
Corrected as suggested Ln 229.
L226 - „i.e., ceca” instead of „called the cecum”
Response
Corrected as suggested Ln 231.
L234 - „gastrointestinal” – GIT
Response
Done as suggested.
L236 - „the most crucial interactions” in the reviewer's point of view one of the most important part of the GIT of birds is the crop which may affect the population in the further parts. However, the Authors omitted/avoided this GIT segment in the manuscript.
KieroÅ„czyk, B., Rawski, M., DÅ‚ugosz, J., ÅšwiÄ…tkiewicz, S., &Józefiak, D. (2016). Avian crop function–a review. Ann. Anim. Sci, 16(3), 653-678.
Classen, H. L., Apajalahti, J., Svihus, B., &Choct, M. (2016). The role of the crop in poultry production. World's Poultry Science Journal, 72(3), 459-472.
Response
We added a paragraph on the importance and role of the crop microbiota Ln 239-254.
L261,275 - „gastrointestinal system” GIT
Response
Corrected as suggested.
L276-278 - „Previous research using germ-free hens showed that the weight and wall thickness of the small intestine and cecum were lower than those of normal chickens.” Please, add citations.
Response
Done as suggested Ln 298.
L302 - „driver” please, do not use a colloquial speech
Response
Corrected as suggested Ln 323.
Figure 2 - this figure does not add any novel information; thus, from the reviewer's point of view should be removed.
Response
Figure 2 summarizes the factors affecting gut microbiota in a simple and informative way to give the reader better experience in reading our review article with different presentations of the information.
L346 - „native microbiota” it means commensal? Please, change
Response
Corrected as suggested Ln 367.
L441 - „firewall” please, do not use a colloquial speech
Response
The concept of “mucosal firewall” is not a colloquial speech and it is regularly used in the scientific journal, please see the below references:
https://doi.org/10.1038/nrgastro.2014.90
https://doi.org/10.1038/s41577-019-0268-7
https://doi.org/10.1038/nri2710
L476, 478, 480l481- in terms of „CD4+CD8−CD25+ and CD4+CD8+CD25+” please, use the upper index as in the original paper [112].
Response
Done as suggested.
L550 - there is a necessity to add the upper indexes.
Response
Done as suggested.
L552- „significantly improved b in all treatment groups” please, correct.
Response
Corrected.
L555 - „Maiorano and his colleagues” -please, change as follow, Maiorano et al.
Response
Done as suggested Ln 580.
L563 - „didn't significantly impact” please, check the grammar
Response
Done as suggested Ln 588.
L592 - „Bi2tos” is it correct?
Response
Yes, this is correct.
Figure 3 - the quality of the figure is absolutely poor, please, correct.
Response
The quality of Figure 3 has been improved.
L649 - „Bifidobacterium” please, use italics
Response
Done as suggested Ln 675.
651, 666 - „and colleagues” - please, change to „et al.”
Response
Done as suggested Ln 677, 692.
658 - „short-chain fatty acids (SCFA)” change to „SCFA”
Response
Done as suggested Ln 684.
665 - „Bifidobacteriaceae and Lactobacillae” facility without the italics
Response
Done as suggested Ln 691.
L672 - potentially pathogenic bacteria
Response
Done as suggested Ln 698.
L679 - „Clostridium” Please use italics
Response
Done as suggested Ln 705.
L680-685 - the sentence have to be removed due to the fact that the numerical changes cannot be taken into consideration (the result is non-significant).
Response
Done as suggested.
L687, 690 - L. plantarum - please, use italics; inulin
Response
Done as suggested Ln 708, 709, 711.
References
- Ramírez, G.A., et al., Broiler chickens and early life programming: Microbiome transplant-induced cecal community dynamics and phenotypic effects. Plos one, 2020. 15(11): p. e0242108.
- Rubio, L.A., Possibilities of early life programming in broiler chickens via intestinal microbiota modulation. Poultry science, 2019. 98(2): p. 695-706.
- Cherian, G., Essential fatty acids and early life programming in meat-type birds. World's Poultry Science Journal, 2011. 67(4): p. 599-614.
- Jha, R., et al., Early nutrition programming (in ovo and post-hatch feeding) as a strategy to modulate gut health of poultry. Frontiers in veterinary science, 2019. 6: p. 82.

Round 2
Reviewer 3 Report
Dear Corresponding Author,
The reviewer wants to thank the Authors to take into consideration all comments and suggestions. Please, find only minor changes which should be done before the final positive recommendation.
Comment 1 - the abstract section is definitely too long. Please, find the information about the number of words in the Guidance for Authors.
Authors response
"The words number in the abstract is only 290 words!!"
In the Instruction for Authors (https://www.mdpi.com/journal/animals/instructions) there is clearly stated that the Abstract section should not exceed 200 words. Thus, in the present version of the manuscript, the Abstract is too long.
Comment 2 - During the correction some added words are not edited correctly, e.g., L80, 83-84, 94, etc. please, correct the whole manuscript in this case.
Comment 3 - There is still a problem with the usage of italics font in the scope of genus and species Latin names, e.g., L253. Please, correct the whole manuscript including Tables.
Author Response
Response letter
Manuscript ID: animals-1462972
Title: Modulation of gut microbiota via in ovo technique mediates early-life programming in broiler chickens
Journal: Animals, MDPI
Corresponding author: Abdelrazeq M. Shehata
Subject: Submission of a revised manuscript
Dear Dr. Antonia Popa,
Managing Editor
Animals, MDPI
Thank you very much for handling and processing our manuscript. According to your comments, we have revised the manuscript extensively. We provide a point-by-point response to the comments in this letter. All the revisions in the manuscript are highlighted in yellow color. We hope that this revised version will be acceptable for publication.
Yours sincerely,
Abdelrazeq M. Shehata
abdelrazeq@azhar.edu.eg
Department of Animal Production,
Faculty of Agriculture,
Al-Azhar University, Cairo, Egypt
Dear Corresponding Author,
The reviewer wants to thank the Authors to take into consideration all comments and suggestions. Please, find only minor changes which should be done before the final positive recommendation.
Thank you very much for your supportive comments and giving us a chance to improve our paper.
Comment 1 - the abstract section is definitely too long. Please, find the information about the number of words in the Guidance for Authors.
Thanks for this comment, done as suggested.
The words number in the abstract is only 290 words!!
Thanks for this comment, done as suggested.
In the Instruction for Authors (https://www.mdpi.com/journal/animals/instructions) there is clearly stated that the Abstract section should not exceed 200 words. Thus, in the present version of the manuscript, the Abstract is too long.
Thanks for this comment, done as suggested (The words number in the abstract is 199).
Comment 2 - During the correction some added words are not edited correctly, e.g., L80, 83-84, 94, etc. please, correct the whole manuscript in this case.
Thanks for this comment, done as suggested in line 72, 75, 86, 106, 108 and 116.
Comment 3 - There is still a problem with the usage of italics font in the scope of genus and species Latin names, e.g., L253. Please, correct the whole manuscript including Tables.
Thanks for this comment, done
This manuscript is a resubmission of an earlier submission. The following is a list of the peer review reports and author responses from that submission.